# Mining of QTLs for Spring Bread Wheat Spike Productivity by Comparing Spring Wheat Cultivars Released in Different Decades of the Last Century

**DOI:** 10.3390/plants13081081

**Published:** 2024-04-12

**Authors:** Natalia Shvachko, Maria Solovyeva, Irina Rozanova, Ilya Kibkalo, Maria Kolesova, Alla Brykova, Anna Andreeva, Evgeny Zuev, Andreas Börner, Elena Khlestkina

**Affiliations:** 1Federal Research Center, N.I. Vavilov All-Russian Institute of Plant Genetic Resources, 190121 St. Petersburg, Russia; maria.soloveva.97@mail.ru (M.S.); i.rozanova@vir.nw.ru (I.R.); i.kibkalo@vir.nw.ru (I.K.); markolesova@yandex.ru (M.K.); a.brykova@vir.nw.ru (A.B.); karandash_85@inbox.ru (A.A.); e.zuev@vir.nw.ru (E.Z.); director@vir.nw.ru (E.K.); 2Leibniz Institute of Plant Genetics and Crop Plant Research (IPK), Gatersleben, Corrensstraße 3, D-06466 Seeland, Germany; boerner@ipk-gatersleben.de

**Keywords:** conservation of genetic diversity, grain quality, GWAS, landraces, spring bread wheat, spike productivity

## Abstract

Genome-wide association studies (GWAS) are among the genetic tools for the mining of genomic loci associated with useful agronomic traits. The study enabled us to find new genetic markers associated with grain yield as well as quality. The sample under study consisted of spring wheat cultivars developed in different decades of the last century. A panel of 186 accessions was evaluated at VIR’s experiment station in Pushkin across a 3-year period of field trials. In total, 24 SNPs associated with six productivity characteristics were revealed. Along with detecting significant markers for each year of the field study, meta-analyses were conducted. Loci associated with useful yield-related agronomic characteristics were detected on chromosomes 4A, 5A, 6A, 6B, and 7B. In addition to previously described regions, novel loci associated with grain yield and quality were identified during the study. We presume that the utilization of contrast cultivars which originated in different breeding periods allowed us to identify new markers associated with useful agronomic characteristics.

## 1. Introduction

The ongoing climate change, growing world population, and increasing food demand require higher yields and adaptability of crops, including spring bread wheat. Modern plant breeding based on achievements of genetics has a significant impact on plant resistance to diseases, climate variability, and other factors. Wheat (*Triticum aestivum* L.) is the most widespread crop in the world, cultivated from the northern polar latitudes to the southernmost parts of Australia, Africa, and America. Wheat accounts for around 30% of the world’s grain production and grain reserve [1,2]. Wheat harvests affect the global economy. Breeders are keen to develop improved cultivars based on phenotypic records of grain yield and quality data, expecting that stable harvests will be maintained under various environmental conditions [3]. The efforts to develop cultivars combining resistance to adverse biotic and abiotic factors with high yield have remained relevant for many years. Wheat is an allohexaploid, with a genome size of 17.2 Gbp [4,5,6]. Numerous studies have been conducted to detect genomic regions associated with morphological characteristics. The quantitative trait loci (QTL) method is often used for such studies. The use of this method helps to detect QTLs on all wheat chromosomes. Most of these studies have employed linkage mapping in biparental populations. This method is currently the most common tool for mapping causal plant genome regions. However, it can only examine the parental alleles, ignoring all other alleles found in the population from which the parents were selected. A genome-wide association study (GWAS), unlike QTLs, is a well-established and useful method for mapping various important morphological and agronomic traits of plants. GWAS is a modern approach to molecular and genetic studies required for such a complex plant as wheat due to the size and structure of its genome [7]. GWAS is a field of biological research that studies associations between genome variants and phenotypic traits [3,8]. Through GWAS, new genetic markers associated with grain yield and quality can be identified, and the association of previously detected loci can be confirmed. GWAS is widely applied in crop investigations to study genetic features related to important agronomic characteristics in plants [9,10,11]. The results of such a study can be used for the rapid breeding and development of new cultivars with desirable traits. It is worthy of note that applying GWAS on plants has its limitations: the technique is sensitive to the presence of a large number of related germplasm accessions that agricultural collections often possess. This means that GWAS may be employed to detect loci when the studied population is as heterogeneous as possible, with a sample size of at least 100. Expanding the sample size for the GWAS method increases the statistical significance of the results. It is also necessary to consider the impact of the environment on trait expression and to study crop accessions in the field for three years [12,13]. Technically, the results obtained from a three-year study are summarized using meta-analysis [14], which allows the results of individual studies to be pooled. Methods of pooling *p*-values from a number of similar studies are often known to help improve the statistical power. Also, the statistical significance will be influenced by the diversity of an accession, which is based on the geography and the selection of cultivars from different originators. GWAS is currently widely used to study various wheat accessions and identify loci associated with such important agronomic characteristics as yield, disease resistance, and nutrient content [15,16,17]. 

In the current study, we applied GWAS to detect QTLs for spike productivity and grain quality (as well as plant height and resistance to lodging) using a panel of spring *T. aestivum* L. cultivars released in different decades of the last century. To compose a GWAS panel as diverse as possible, we used cultivars released before and after the “green revolution”. The divergence between old and modern spring wheat cultivars (i.e., the one related to spike productivity and grain quality) released in Russia and Germany was revealed earlier [18,19]. Therefore, we chose Russian and German spring wheat cultivars developed in different decades of the last century. We hypothesized that such an approach to sampling would allow us not only to confirm known loci among the sample, but also to identify new ones. Overall, studying the genetic bases of grain yield and quality by comparing wheat cultivars developed in different periods, including landraces preserved as sources of valuable genetic diversity for breeding and genetic studies, makes it possible to understand selection processes of the past and develop strategies for future breeding programs. 

## 2. Results

### 2.1. Phenotyping

In this study, 186 accessions were evaluated at VIR’s experiment site (latitude: 59.71482042142053; longitude: 30.42364618465661) across 3 years of field trials. Phenotyping was carried out according to VIR’s standard technique [20]. The weather conditions of the experiment varied from year to year, which affected the phenotypes of the accessions. The shortest germination-to-heading period was in 2021. Greater plant heights were observed in 2022, and lower heights in 2023. The length of the spike, the number of spikelets per spike, and the number of grains per spike were minimal in the drier year of 2023. The grain weight per spike, the weight of 1000 grains, and the yield were higher in 2022 and lower in 2023 (Table 1).

The phenotypic statistics (minimum, maximum, and mean) for five characteristics are presented in Table 1.

The panel of spring bread wheat cultivars included genotypes with high variability in useful agronomic characteristics (for example, SL varied from 3.3 to 11.6, and TGW from 14.7 to 61.7) (Table 1). At the same time, the presence of approximately the same range of values for each character in each accession during a three-year period allowed us to assess the contribution of the genetic component to the development of each characteristic.

### 2.2. Analysis of Genotyping Data

The studied spring bread wheat accessions were heterogeneous. Using the Structure software v 2.3.4, they were divided into six groups, or subpopulations (k = 6) (Figure 1).

### 2.3. Association Analysis

The obtained phenotype and genotype data were used for association analysis. The results were first compared in the QQ-plot. Then, the Manhattan plots were designed (Figure 2, Appendix A).

#### 2.3.1. Productivity

In total, 24 SNPs associated with the PH, RL, SL, TGW, and SN characteristics were identified (Figure 2, Appendix A, Table 2).

In addition to detecting significant markers for each year of the field study, meta- analyses were conducted. Significant markers were identified for all three years of the study. As a result, one locus associated with PH was revealed on chromosome 5A (585-609 Mb). This locus was observed for two years: 2022 and 2023. Only one locus was found for TGW (chromosome 4A, 684 Mb) and SN (chromosome 7B, 648 Mb). These loci were identified by a meta-analysis and had suggestive levels. In total, two loci associated with SL were identified on chromosome 6A (410 Mb) and chromosome 6B (470 Mb), also by a meta-analysis. Five suggestive loci associated with RL were detected on chromosomes 2A (779 Mb), 2B (448 Mb), 2D (634 Mb), 6A (613 Mb), and 7A (736 Mb). 

#### 2.3.2. Technological Evaluation

The technological properties of the grain of spring bread wheat accessions were assessed involving the following indicators:(1)Protein content is one of the main indicators of the nutritional value of grain. It is considered that this parameter does not depend directly on gluten quality. In some cases, its antagonism is observed in relation to grain quality indicators associated with gluten properties.(2)Ash content also characterizes the nutritional and forage value of grain. Its increased values are regarded as an unfavorable factor that reduces the nutritional value.(3)Test weight is one of the important criteria for assessing technological properties of wheat grain. It has a direct impact on flour yield during grain milling. Moreover, this indicator is not directly linked to gluten or starch quality, which makes it an independent characteristic in the genetic sense. Test weight can be affected by grain size and its density.(4)Flour color. This indicator was assessed in whole-grain flour; therefore, its intensity, aside from the color of endosperm particles, depended on the ratio of grain hulls to endosperm. The higher its value, the more this ratio was shifted towards endosperm.(5)Sedimentation value. The sedimentation method is widely used in grain quality evaluation. The SDS-sedimentation techniques have received most recognition in breeding and genetic research for the large-scale evaluation of breeding material. Sedimentation value is considered to be a generalized criterion of grain quality that characterizes the capacity of ground grain for stable swelling, as well as the quality of storage proteins and gluten. Such an approach to grain quality evaluation is highly efficient and involves low material consumption. Sedimentation value was noted to have a high level of heritability in progenies compared to other criteria of grain quality.

All measured indicators of grain technological properties had high intercultivar variability, while the accessions had significant F-criterion differences (Table 3).

The highest “Test weight” values were registered in three Russian cultivars released in different decades of the last century (VIR catalog Nos. k-632855, k-632897 and k-632898) and a German landrace (k-632980); the lowest values were found in a number of German cultivars released before 1950 (k-632922, k-632923, k-632948 and k-632949) and one Russian cultivar (k-632920).

The highest “Protein content” in grain was recorded for several Russian cultivars released in different decades of the last century (k-632852, k-632864, k-632879 and k-632920), and German ones released before 1950 (k-632921, k-632922 and k-632923); the lowest values of this indicator were observed in several Russian and German cultivars developed in different time periods.

The highest “Sedimentation value” was demonstrated by three Russian (k-632871, k-632876 and k-632887) and three German cultivars (k-632949, k-632995 and k-633006) released in different decades of the last century; the lowest was found in three German cultivars released in 1951–1991 (k-633007, k-633009 and k-633010).

The lowest levels of “Ash content” in grain were found in several Russian cultivars developed in different decades of the last century, as well as landraces; the same group included two German cultivars released before 1950 (k-632938 and k-632943). The highest value of this indicator was registered in one Russian accession (k-632920) and two German ones (k-632922 and k-633008).

The lightest flour color was observed in a number of Russian cultivars developed in different periods and a German cultivar released in the second half of the 20th century (k-632983). Russian and German cultivars dating back to early periods of breeding were characterized by the darkest flour color.

Thus, judging by the phenotypic manifestation of the studied grain quality indicators, valuable accessions for breeding were identified in both the Russian and German groups of cultivars developed in all time periods under consideration. Accessions with negative manifestations of important agronomic characteristics were often found among the cultivars released in early periods of breeding (Appendix A).

Based on the genotyping and evaluation results for grain quality indicators, GWAS was carried out for wheat accessions studied in 2022 under field conditions at the experimental site of VIR. 

Loci associated with yield-related agronomic characteristics were detected on chromosomes 4A, 5A, 6A, 6B, and 7B. Both significant and suggestive markers were identified (Table 4).

A significant locus associated with flour color was identified on chromosome 5A (468–473 Mb), and one locus (suggestive level) was found on chromosome 2B (441 Mb). 

Nine suggestive markers according to the grain test weight indicator and five significant markers according to the protein content indicator were identified. Four markers for ash-content, nine markers associated with flour color, and eighteen markers associated with flour sedimentation were detected. The detected markers were found on chromosomes 1A–7A, 1B–7B, 1D, 2D, and 6D.

Thus, loci associated with grain yield and quality, as well as individual markers widely covering bread wheat chromosomes, were identified during the study.

## 3. Discussion

In a number of studies, including those on wheat [21,22,23,24], a comparison among cultivars released in different decades for the same cultivation area was conducted using the DNA polymorphism analysis. The data obtained in these studies clearly demonstrated that, in a certain period of cultivation, there is a certain proportion of unique alleles that are no longer found in the next generation of cultivars developed for the same cultivation area. These genome-based data not only confirmed the need to preserve already existing ex situ collections and regularly replenish them, but also pointed to out-of-use cultivars as an important potential source of genetic diversity for breeding new cultivars. Therefore, cultivars released in different periods and landraces, along with modern cultivars, are widely included in panels used for comparative genetic studies [20,25,26]. This enables researchers not only to find sources of valuable traits among ex situ collection accessions (including sources for improving grain and flour quality among old cultivars [19,27,28]), but also to identify genome loci containing potentially valuable alleles. In turn, this makes it possible to develop technologies for accelerated marker selection and to discover new target genes for editing [29,30]. 

Many studies investigating the global and local wheat genetic diversity, aimed at identifying donors, have generally ignored landraces adapted to local environmental conditions. In our work, the sample under study was formed of spring wheat cultivars bred in different decades of the last century in Russia and Germany (Table 5), which allowed us to identify new markers associated with useful agronomic characteristics. 

Spike length, number of spikelets per spike, thousand-grain weight, plant height, and resistance to lodging are the characteristics that, together, determine the level of wheat yield. These characteristics correlate with each other, and by identifying a locus associated with one of them, we can see that it affects other characteristics as well. In this study, we identified a locus on chromosome 5AL associated with plant height and resistance to lodging (122.1–122.6 cM). It was located on chromosome 5A in the physical interval of 585.0–588.0 Mb, next to those previously discovered [31,32,33]. Also, four markers associated with resistance to lodging were identified on chromosomes 2A, 2B, 2D, 6A, and 7A. Spike length and the number of spikelets per spike, as shown in the previous studies, are controlled by several genes, the main ones of which are located on chromosomes 5A, 2D, and 3D [34,35]. However, QTLs have been identified on almost all wheat chromosomes [36,37,38].

Using the meta-analysis, we identified two markers associated with spike length: the marker (RAC875_c48456_444) on chromosome 6B and the marker of a suggestive level (Excalibur_rep_c92855_977) on chromosome 6A. Also, a suggestive level marker associated with the number of spikelets per spike was revealed on chromosome 7B (AX-94505099). As has been shown in previous studies, many major QTLs related to thousand-grain weight have been mapped to almost all wheat chromosomes [36,39,40,41,42,43,44,45,46,47,48]. We identified one marker associated with thousand-grain weight (Excalibur_c4325_1150) on chromosome 4A.

A large number of studies have searched for genes associated with high protein content in grain. Thus, the *NAM-B1* (*Gpc-B1*) gene is known, and has been identified on wheat chromosome 1B [49]. Other candidate genes associated with grain protein content have also been identified, such as *QGpc.ipk-7B* [50], *QGlc.ipk-5B*, and *QGlc.ipk-7A* [51]. These genes are localized on chromosomes 7B and 7A, respectively. In this study, we found new loci on chromosomes 1A, 1D, 2A, and 2D. Other technological characteristics of wheat grain at the molecular level are also being studied, the color of flour being one of them. It is characteristic of wheat cultivars that produce a significant amount of carotenoids. Discoloration of flour may be induced by high activity of the enzyme lipoxygenase [52]. Lipoxygenase activity is associated with a locus on wheat chromosome 4BS, which has been well studied at the molecular level [53,54]. In addition to chromosome 4B, loci associated with this indicator were identified on chromosomes 7A (QTL/7AL) [55,56] and 1B (*Psy-B1* and *Lpx-B1*) [57,58]. We discovered a locus on chromosome 5A and a marker on chromosome 2B associated with flour color.

In our opinion, such an approach to choosing contrast genotypes could be one of the reasons why we were able to find some new genomic loci significantly associated with the evaluated characteristics.

Thus, in this study, we obtained new information and identified new loci associated with the indicators of wheat yield and grain quality. These studies should be continued to identify new candidate genes for their further use in the development of new wheat cultivars.

## 4. Materials and Methods 

### 4.1. Plant Material and Genotyping Data

The panel of cultivars was composed taking into account their genetic and phenotypic diversity: 186 accessions in total, including those received by VIR from the IPK Gatersleben bread wheat collection (IPK), belonging to 10 bread wheat varieties (*lutescens* (Alef.) Mansf., *ferrugineum* (Alef.) Mansf., *milturum* (Alef.) Mansf., *erythrospermum* Korn., *albidum* Al., *caesium* (Alef.) Mansf., *graecum* (Koern.) Mansf., *fulvocinereum* Flaksb., *subferrugineum* (Vav.) Mansf., and *pyrothrix* (Alef.) Mansf.). The selected set of accessions consisted of cultivars of Russian origin from 24 regions (51%) and those originating from breeding centers in Germany (49%). Landraces, old cultivars, and modern cultivars were among the studied wheats (Table 5). The names of the botanical varieties are given according to the classification adopted at VIR [59]. 

### 4.2. Field Experiment and Phenotyping

The experimental design of each trial was completed with three replications during 2021–2023 in the fields of the research and production facility of VIR in Pushkin, St. Petersburg, Russia (latitude: 59.71482042142053; longitude: 30.42364618465661). The soil of the site was light loam. The wheat collection was studied in accordance with the guidelines published earlier [20]. Each cultivar was grown on a 1 m^2^ plot. The reference cultivars were “Leningradskaya 97” and “Leningradskaya 6”. Mineral fertilizers (N30, P50, and K30) were applied annually before sowing. Chemical treatments were applied against the Swedish fly with the drug Danadin. Sowing and harvesting were carried out manually. The sheaves were threshed using a Wintersteiger LD 350 thresher. During threshing, 10 typical spikes were randomly taken from each accession for analysis. 

The spring of 2021 was cool, with twice the normal amount of precipitation falling in May, which led to flooding of some plots. In June/July, a deficit of precipitation was observed. Most spring wheat accessions ripened earlier. 

May 2022 was very dry, with only 4.3 mm of rainfall (normal rate is 46 mm). The summer of 2022 manifested average temperatures in terms of temperature values, and only July was a little cool. In August, the rainfall was almost twice as high as the norm, which slightly lengthened the growing season of late wheat accessions. Overall, the weather conditions in 2022 were the most favorable for wheat.

The spring of 2023 had the following features: in March, the rainfall was three times higher than the monthly norm, and in May, on the contrary, there was a drought—only 7.1 mm of precipitation was recorded. However, the June rains had a positive effect on the further growth of wheat. In terms of temperature, July was cool, and August was at the level of long-term average values. September was warm. 

### 4.3. DNA Isolation

DNA isolation was performed with the DNeasy Plant Mini Kit (Qiagen) [60] under the manufacturer’s protocol. The measurement of DNA concentration was carried out using the NanoPhotometer NanoDrop.

### 4.4. Sample Genotyping

The sample was genotyped using the array 20K Wheat Illumina SNP chip containing 17,267 polymorphic SNPs by TraitGenetics GmbH (www.traitgenetics.com (accessed on 8 April 2024)). This array is an optimized and reduced 15K version [61], and 5385 markers from the 35K Wheat Breeders Array [62] were added. SNPs with minor allele frequency (MAF) ≤ 0.05 and missing values > 5% were removed from the subsequent analysis, leaving a set of 13,375 polymorphic SNP markers. 

### 4.5. Population Structure

The accessions were clustered using the STRUCTURE v 2.3.4 software package [63].

### 4.6. Statistical Analysis

The statistical analysis (except GWAS) for all characteristics was conducted according to Dospekhov [64]. STATISTICA 10.0 software was employed.

### 4.7. Association Analysis

Association analysis was performed with the TASSEL 5 software [14] using the following formula: Y = Xa + Qb + e, where Y is a vector for the phenotypic values, X is a matrix of the genotypic values of the marker, a is the vector of fixed effects of the marker, Q is the kinship matrix, b is a vector of the fixed effects of population structure, and e is a vector of the residual effects. We used a mixed linear model (MLM) that accounts for a genetic relationship among cultivars and the sample structure. 

To detect significant SNPs, we employed the Bonferroni threshold of 3.74 × 10^−6^, based on the significance level (0.05) divided by the number of effective markers (13,375). Thus, we used -log10 (*p*-value) = 5.43 as the significance threshold for different indicators in our study. We selected a robust model to account for population structure and believed that it would already account for most false positives. We therefore set an estimated threshold of *p* < 10^−4^ to highlight candidates that were small enough, but did not reach the significance level. 

### 4.8. Meta-Analysis

A meta-analysis was used to enhance the statistical power of the GWAS results [65]. Therefore, we also combined the *p*-values of three years for each studied character. We used the modified Fisher’s method, accounting for the non-independence implemented in the R package “poolr”, because the results were not independent (each year, we took the same genotypes) [66]. It exploited the effective rather than nominal number of tests, thus avoiding overestimation of statistical significance. The effective number of tests was calculated using the Nyholt method [67].

### 4.9. Technological Evaluation of Grain 

The test weight was measured using grain test weight micro scales in accordance with the guidelines by Vasilenko and Komarov [68]. The metal chamber of the test weight micro scales was evenly filled with grain and compacted; then, a 10 mL volume was cut off using a special knife. Surplus grain was removed, while grain remaining in the 10 mL chamber was weighed to within 0.01. The resulting weight was multiplied by 100. The study included the sedimentation techniques devised by Bebyakin and Buntina for spring bread wheat [69].

Microsedimentation of 0.5 g of whole-grain flour obtained on a cyclone mill was placed into a 10 mL measuring tube, then 4 mL of distilled water was added and shaken intensively. Then, the sample was suspended on a rotator for 2 min. After that, 6 mL of the working solution was added (17% SDS, 3% acetic acid) and suspended again on a rotator for 5 min. Finally, sedimentation was carried out for 15 min, and the sediment volume was measured and multiplied by 10 [69].

Protein content, ash content, and whole-grain flour color were identified indirectly using the Perten IM 9500 IR spectrometer according to the instructions for the device.

## 5. Conclusions

The results of the field study (phenotyping) of 186 spring wheat accessions, conducted in 2021–2023 with two-fold repetition at the research and production facility of VIR, are summarized herein. The accessions were evaluated according to yield-related characteristics, such as plant height (PL), resistance to lodging (RL), spike length (SL), and thousand-grain weight (TGW). Loci associated with useful agronomic characteristics were detected through GWAS. The results of the study highlight the importance of conserving the gene pool of landraces as a source of valuable genetic diversity for breeding and genetic research.

## Figures and Tables

**Figure 1 plants-13-01081-f001:**
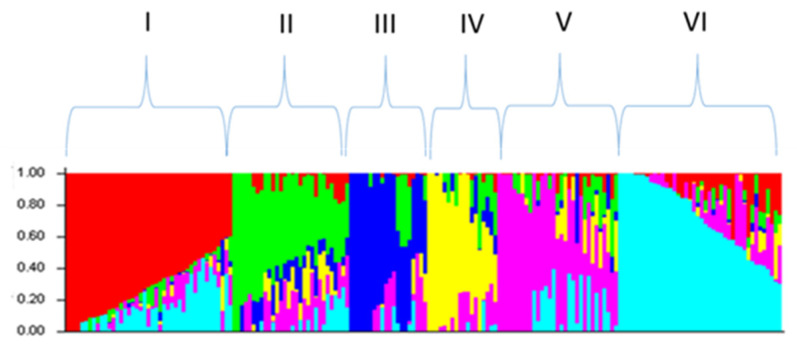
Subgrouping of wheat accessions based on 17,267 SNP markers. Delta K value (k = 6). I–VI—subgroup numbers.

**Figure 2 plants-13-01081-f002:**
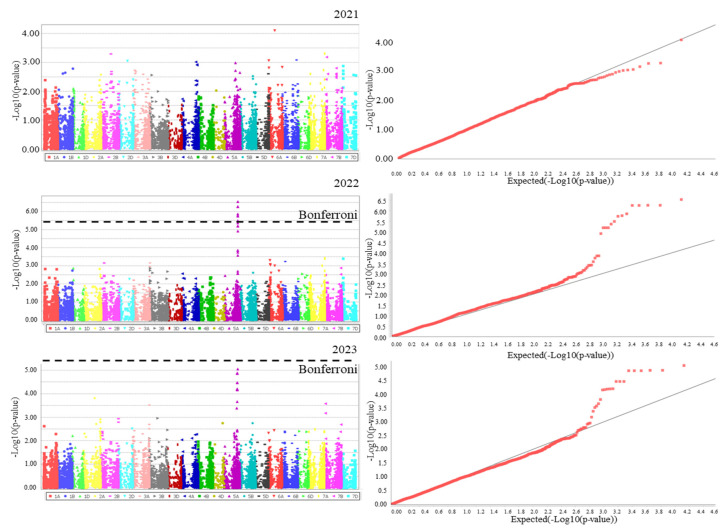
Manhattan plots and QQ-plots of the PH agronomic character. The dotted lines in the Manhattan plots indicate the Bonferroni threshold; the red dots and straight gray lines in the QQ-plots indicate the experimental and expected values, respectively.

**Table 1 plants-13-01081-t001:** Descriptive statistics of the studied phenotypic indicators.

Characteristic	Mean for Each of the Three Years			Max			Min		
	2021	2022	2023	2021	2022	2023	2021	2022	2023
Resistance to lodging (RL)	8.0 ± 1.6	7 ± 2.3	7.9 ± 1.4	9	9	9	1	3	3
Spike length (SL)	8.4 ± 1.0	8.3 ± 1.2	7.1 ± 1.0	11.6	11.2	9.5	4.7	4.1	3.3
Plant height (PH)	92 ± 11	103 ± 14.7	86.1 ± 14.9	110	140	125	60	73	52.5
Thousand grain weight (TGW)	33.1 ± 5.6	45.9 ± 6.8	31.4 ± 7.2	46.6	61.7	49.4	20	23.1	14.7
Number of spikelets per spike (SN)	14.7 ± 1.5	15 ± 2.0	13.3 ± 2.1	19.2	22	20.2	11	9	8

**Table 2 plants-13-01081-t002:** Summary of association mapping results for agronomic characteristics.

Characteristic	Marker	Chr	Position	*p*-Value
PH	RAC875_rep_c113313_607^meta^	5A	585240573	2.74 × 10^−6^
	wsnp_Ex_c31799_40545478^2022,meta^	5A	585403218	5.3 × 10^−7^
	wsnp_Ex_c31799_40545376^meta^	5A	585403320	1.08 × 10^−6^
	Excalibur_c7729_144^meta^	5A	585412831	1.05 × 10^−7^
	tplb0038h19_1394^2022,meta^	5A	585431093	1.05 × 10^−6^
	RAC875_c9984_1003^meta^	5A	585458474	8.87 × 10^−7^
	wsnp_Ex_rep_c66689_65010988^2022,meta^	5A	585609287	9.26 × 10^−8^
	BS00022071_51^2022,meta^	5A	586604587	9.46 × 10^−8^
	TG0052^meta^	5A	587412057	1.08 × 10^−6^
	TG0053^meta^	5A	587412186	1.25 × 10^−6^
	TG0019^2022,meta^	5A	587423597	9.74 × 10^−7^
	TG0041^2022,meta^	5A	588550278	5.97 × 10^−8^
	wsnp_BF293620A_Ta_2_1^2022,meta^	5A	588555309	1.04 × 10^−7^
	TA001896-0654^meta^ *	5A	588848205	3.49 × 10^−6^
	AX-94920711^2023^ *	5A	609276661	9.01 × 10^−6^
RL	wsnp_CAP11_rep_c4105_1940985^2021^ *	2B	448080584	1.91 × 10^−5^
	tplb0050d17_1401^2021^ *	6A	613770166	3.87 × 10^−5^
	Tdurum_contig45618_1089^2023^ *	7A	736690246	9.18 × 10^−6^
	BS00024643_51^2023^ *	2A	779207402	4.98 × 10^−5^
	Excalibur_c16329_493^meta^ *	2D	634296660	8.07 × 10^−5^
SL	RAC875_c48456_444^meta^	6B	470800981	1.21 × 10^−6^
	Excalibur_rep_c92855_977^meta^ *	6A	410914096	3.8 × 10^−6^
TGW	Excalibur_c4325_1150^meta^ *	4A	684616475	2.38 × 10^−5^
SN	AX-94505099^meta^ *	7B	648926257	6.31 × 10^−6^

The *p*-value column indicates the smallest value obtained. * Suggestive (values are low enough, but not exceeding the threshold) physical positions were determined from the data source, International Wheat Genome Sequencing Consortium (https://www.ebi.ac.uk/ena/browser/view/GCA_900519105.1 (accessed on 8 April 2024)).

**Table 3 plants-13-01081-t003:** Structure of spring bread wheat accessions according to technological properties of their grain.

Structural Indicators	Indicators				
	Protein Content, %	Ash Content, %	Flour Color, %	Test Weight, g/L	Sedimentation Value, mL
Limits of variation	11.5–20.7	1.53–2.80	74.8–84.1	622–832	16–82
Experiment mean value	14.79	1.99	81.27	760.27	61.49
F-criterion (intercultivar)	3.51 *	3.19 *	1.76 *	2.75 *	3.62 *
HCP	2.22	0.27	3.83	45.28	13.44

*—F-criterion significance.

**Table 4 plants-13-01081-t004:** Association mapping results for grain quality.

Indicator	Marker	Chr	Position	*p*-Value
Test weight	Excalibur_c82557_201 *	1A	9123021	7.20 × 10^−5^
	BS00009789_51 *	5A	451478823	2.85 × 10^−5^
	BobWhite_c8202_245 *	5A	445191670	9.29 × 10^−5^
	IAAV8870 *	5B	473114741	1.64 × 10^−5^
	AX-94541836 *	5B	572140495	6.59 × 10^−5^
	BobWhite_rep_c48956_706 *	6A	149925808	8.25 × 10^−5^
	IAAV8065 *	6B	411097830	8.22 × 10^−5^
	RAC875_c17185_90 *	7A	20164436	6.49 × 10^−5^
	Kukri_c49828_316 *	7B	702501105	6.77 × 10^−5^
Grain protein content	IAAV5730	1A	344480854	5.00 × 10^−6^
	TA004690-1102	1D	435801686	3.33 × 10^−6^
	AX-94602991	2A	776022491	3.28 × 10^−6^
	IACX8602	2A	776040004	3.33 × 10^−6^
	JD_c63957_1176 *	2D	20769330	2.20 × 10^−5^
Ash content	AX-94726440 *	3A	197860384	6.66 × 10^−6^
	BS00065543_51 *	5B	17575036	7.19 × 10^−6^
	AX-94519170 *	6D	464735570	4.00 × 10^−6^
	RAC875_c17185_90 *	7A	20164436	1.25 × 10^−5^
Flour color	Kukri_c57491_156 *	2B	440825097	4.34 × 10^−6^
	wsnp_Ex_c19647_28632894	5A	470033197	1.87 × 10^−6^
	wsnp_JD_c6160_7327405	5A	472344585	1.87 × 10^−6^
	RFL_Contig2187_1025	5A	472346644	1.87 × 10^−6^
	IACX12578	5A	467379740	2.71 × 10^−6^
	BobWhite_c46338_76	5A	468462719	2.71 × 10^−6^
	Kukri_c17430_972	5A	468467336	2.71 × 10^−6^
	AX-94436930 *	5A	473312305	5.69 × 10^−6^
	RAC875_c79944_269 *	5A	468463193	7.50 × 10^−6^
Flour sedimentation	Kukri_c9898_1766	0	0	2.91 × 10^−8^
	AX-94881376	1A	30136011	3.78 × 10^−8^
	wsnp_BF474340A_Ta_2_1	1A	556942097	4.63 × 10^−8^
	IAAV5776	1B	675560975	3.13 × 10^−6^
	AX-94414376 *	1B	552777509	6.20 × 10^−6^
	AX-95213897 *	2A	510805288	9.11 × 10^−6^
	Kukri_c63797_354	3B	761853919	1.89 × 10^−8^
	AX-94467468 *	4A	599326520	9.08 × 10^−6^
	Tdurum_contig8028_870 *	4B	586069506	5.78 × 10^−6^
	wsnp_Ku_c23772_33711538	5A	476603824	4.11 × 10^−8^
	RAC875_rep_c109969_119	5A	593332300	3.40 × 10^−7^
	RAC875_c2105_740	5B	555011247	3.68 × 10^−8^
	Kukri_c13224_551	5B	87230041	3.95 × 10^−8^
	AX-94878420	5B	449201643	4.28 × 10^−8^

*: Suggestive (values are low enough, but not exceeding the threshold) physical positions were determined from the data source, International Wheat Genome Sequencing Consortium (https://www.ebi.ac.uk/ena/browser/view/GCA_900519105.1 (accessed on 8 April 2024)).

**Table 5 plants-13-01081-t005:** Status of the studied wheat accessions.

Accession Status	Number of Wheat Accessions	
	From Russia	From Germany
Landraces	19	10
Cultivars released before 1950	19	51
Cultivars released in 1951–1991	42	30
Modern improved cultivars	14	1
Total	94	92

The weather conditions of the experiment varied from year to year.

## Data Availability

Data are contained within the article and Appendix A.

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
