# Peer review of "Mining of QTLs for Spring Bread Wheat Spike Productivity by Comparing Spring Wheat Cultivars Released in Different Decades of the Last Century"

_plants, 2024, doi:10.3390/plants13081081_

Round 1

Reviewer 1 Report

Comments and Suggestions for Authors

I have carefully read the article. I believe that the data and results presented in this article hold some significance. However, I regret to say that I find the writing quality to be rather mediocre and not up to the standard required for publication. The abstract lacks any data and is insufficient for understanding the content. There are numerous grammatical errors throughout the article. The Manhattan plot is unclear, and the tables lack unit information among other details. The results section contains many unnecessary details. The background section barely introduces the traits and focuses mainly on the association analysis. In my opinion, significant improvements in writing quality are necessary for this article to be considered for publication.

Comments on the Quality of English Language

English need extensive editing.

Author Response

Dear reviewer, thank you for deeply and carefully reading our article.

Below you can find our answers:

I have carefully read the article. I believe that the data and results presented in this article hold some significance. However, I regret to say that I find the writing quality to be rather mediocre and not up to the standard required for publication. The abstract lacks any data and is insufficient for understanding the content. There are numerous grammatical errors throughout the article. The Manhattan plot is unclear, and the tables lack unit information among other details. The results section contains many unnecessary details. The background section barely introduces the traits and focuses mainly on the association analysis. In my opinion, significant improvements in writing quality are necessary for this article to be considered for publication.

In my opinion, the manuscript solves an interesting issue, but in the current version there are several problematic things that need to be revised.

The abstract is written very generally, it should be supplemented with concrete results achieved.

- The abstract lacks any data and is insufficient for understanding the content.

 – We corrected the abstract

Abstract: Genome-wide association studies (GWAS) are among the genetic tools for the mining of genomic loci associated with useful agronomic traits. The study enabled us to find new genetic markers associated with grain yield as well as quality. The sample under study consisted of spring wheat cultivars developed in different decades of the last century. A panel of 186 accessions was evaluated at VIR’s experiment station in Pushkin across a 3-year period of field trials. In total, 24 SNPs associated with six productivity characters were revealed. Along with detecting significant markers for each year of the field study, meta-analyses were conducted. Loci associated with useful yield-related agronomic characters were detected on chromosomes 4А, 5А, 6А, 6В, and 7В. In addition to previously described regions, novel loci associated with grain yield and quality were identified during the study. We presume that utilization of contrast cultivars originated in different breeding periods allowed us to identify new markers associated with useful agronomic characters.

- There are numerous grammatical errors throughout the article. –

- The MS text was revised for English grammar.

- The Manhattan plot is unclear, and the tables lack unit information among other details.

- The figures were improved, supplied with detailed information and presented as supplementary file.

- The results section contains many unnecessary details.

- We moved some details into another paragraphs (for example, agroclimatic conditions in “Materials”) or in supplementary data.

  1. Results - 2.1. Phenotyping - The weather conditions of the experiment varied from year to year. In my opinion, the agroclimatic conditions in each year of the experiment (2021, 2022, 2023) should be listed here and then compared for individual characteristics (protein content, etc.).

- Weather conditions of the experiment are described by year of study (4.2). We have added information about the influence of weather conditions on certain agronomic characteristics (2.1)

In this version of the manuscript, I miss the comparison of the effect of different breeding periods and programs on the analyzed genotypes from Germany and from Russia.

- The main idea to use cultivars released in different periods (namely that released before and after the “green revolution”) was rather to compose a GWAS panel as diverse as possible, than to focus on comparison of old and modern varieties. As for different breeding programs we have chosen Russian and German varieties based on previous data on divergence between of old and modern spring wheat varieties (namely, that related with spike productivity and grain quality) released in these countries (Hagel et al., 2005; Morozova et al., 2016, etc). In the revised MS we do not focus anymore in the article title on different breeding programs, however, we explain our considerations when compiling the GWAS panel. In electronic supplementary materials for the revised MS we included lists of cultivars with low and high values for different spike productivity and grain quality traits, from which one can see that for some traits, cultivars released in different periods contribute to subgroups with either highest or lowest values. We think, such way to choose contrast genotypes may be one of the reason we were able to find some new genomic loci significantly associated with evaluated traits. We discuss this issue in the revised MS.

2.3.2. Technological evaluation – In this part, it is necessary to better describe the achieved results (table 3) in relation to individual years of the experiment and different genotypes (Russia, Germany).

- In the current study we focused on genetic analysis and GWAS. The purpose of this article was to demonstrate whether it is fundamentally possible to identify loci (including new ones), associated with grain yield as well quality properties through analysis using the novel GWAS panel consisting of varieties released in Russia and Germany before and after the “green revolution”. The results obtained by GWAS inspired us to carry out further extended research not only at one location in Pushkin, but also additional three-year studies at several more experimental stations (in different geographical regions). The purpose of further expanded research will also be a more detailed comparative presentation of phenotypic characteristics, while the goal of the current study was mainly genotyping, analysis of population structure and GWAS in the main experimental station. But despite this, taking into account this reviewer's comment, we however decided to add more descriptive data on the phenotype (see appendix 2).

4.2. Field experiment and phenotyping: Why is only 2023 characterized here? It is necessary to characterize the acroclimatic conditions in each experimental year (weather course + soil nutrient supply)

- Weather conditions of the experiment are described by year of study (4.2).

- We have added this information in the revised MS. The site soil is light loam; mineral fertilizers (N30, P50, K30) are applied annually before sowing

2.1. Phenotyping

- In this study, 186 accessions were evaluated at VIR’s experiment site (Latitude: 59.71482042142053| Longitude: 30.42364618465661) across 3 years of field trials. Phenotyping was carried out according to VIR’s standard technique [20]. The weather conditions of the experiment varied from year to year, which affected the phenotypes of the accessions. The shortest germination-to-heading period was in 2021. Greater plant heights were observed in 2022, and lower heights in 2023. The length of the spike, the number of spikelets per spike, and the number of grains per spike were minimal in the drier year 2023. The grain weight per spike, the weight of 1000 grains, and the yield were higher in 2022 and lower in 2023 (Table 1).

Table 1. Descriptive statistics of studied phenotypic traits.

Character

Mean in each of the three years

Max

Min

2021

2022

2023

2021

2022

2023

2021

2022

2023

Resistance to lodging

(RL)

8.0 ± 1.6

7 ± 2.3

7.9 ± 1.4

9

9

9

1

3

3

Spike length

(SL)

8.4 ± 1.0

8.3 ± 1.2

7.1 ± 1.0

11.6

11.2

9.5

4.7

4.1

3.3

Рlant height

(PH)

92 ± 11

103 ± 14.7

86.1 ± 14.9

110

140

125

60

73

52.5

Thousand grain weight

(TGW)

33.1 ± 5.6

45.9 ± 6.8

31.4 ± 7.2

46.6

61.7

49.4

20

23.1

14.7

Number of spikelets per spike

(SN)

14.7 ± 1.5

15 ± 2.0

13.3 ± 2.1

19.2

22

20.2

11

9

8

The panel of spring bread wheat cultivars included genotypes with high variability in useful agronomic characters (for example, SL varied from 3.3 to 11.6, and TGW from 14.7 up to 61.7) (Table 1). At the same time, the presence of approximately the same range of values for each character in each accession during three years allowed us to assess the contribution of the genetic component to the development of each character.

4.2. Field experiment and phenotyping:

The experimental design of each trial was completed with three replications during 2021–2023 in the fields of the research and production facility of VIR in Pushkin, St. Petersburg, Russia (Latitude: 59.71482042142053| Longitude: 30.42364618465661). The soil of the site was light loam. The wheat collection was studied in accordance with the earlier published guidelines [20]. Each cultivar was grown on a 1 m2 plot. The reference cultivars were ‘Leningradskaya 97’ and ‘Leningradskaya 6’. Mineral fertilizers (N30, P50, and K30) were applied annually before sowing. Chemical treatments were applied against the Swedish fly with the drug Danadin. Sowing and harvesting were done manually. The sheaves were threshed using a Wintersteiger LD 350 thresher. During threshing, 10 typical spikes were randomly taken from each accession for the analysis.

The spring of 2021 was cool, with twice the normal amount of precipitation falling in May, which led to flooding of some plots. In June/July, a deficit of precipitation was observed. Most spring wheat accessions ripened earlier.

May 2022 was very dry, with only 4.3 mm of rainfall (normal rate is 46 mm). The summer of 2022 manifested average temperatures in terms of temperature values, only July was a little cool. In August, the rainfall was almost twice higher than the norm, which slightly lengthened the growing season of late wheat accessions. Overall, the weather conditions in 2022 were the most favorable for wheat.

The spring of 2023 had the following features: in March, the rainfall was thrice higher than the monthly norm, and in May, on the contrary, there was a drought – only 7.1 mm of precipitation was recorded. However, the June rains had a positive effect on the further growth of wheat. In terms of temperature, July was cool, and August was at the level of long-term average values. September was warm.

Edit the Reference section, i.e. unify according to journal requirements.

We have edited this in the revised MS.

The English grammar, style and syntax used in the manuscript were poor and need to be completely revised.

- English grammar has been improved.

Reviewer 2 Report

Comments and Suggestions for Authors

The manuscript presented by Shvachko et al., aimed to identify genomic regions associated with traits of agronomic interest in a panel of bread wheat genotypes. Therefore, the subject of the paper is relevant, it falls in the general scope of the journal and could be of interest to a wide range of researchers engaged in wheat genetic studies and/or varietal development. However, in this version, the study cannot be published in a high-impact scientific journal.

The title of the manuscript suggested an analysis of the effect of different breeding periods and programs (Russia and Germany) which were not discussed.

The Abstract in this version did not adequately summarize the manuscript; too generic, and without any result.

In the introduction section, the issue addressed by the study was insufficiently exposed. It is important to clearly define and appropriately frame the study question. 

The experimental methodology used was not appropriate to support the hypothesis proposed by the authors and, the methods described are insufficient to repeat the results. GWAS analysis should be conducted with BLUP data, to account for genotype-by-year interaction. The authors implied that they had conducted a meta-QTL analysis, however it is not well understood how it was conducted. No mention was made in the materials and methods section. 

The phenotypic traits considered in the study were not uniquely identified. In the text "Number of spikelets in a spike, SN" is confusing with "grain number in a spike, SN"; or thousand grain weight” with "grain weight per spike". The methodology and protocols for determining phenotypic traits were not detailed. 

The results were not clearly explained, and in some cases, they were not preceded by an appropriate discussion in the methods section (i.e. meta-analysis).

In the discussion section, the authors should discuss in detail the original results (if any) obtained from the GWAS, comparing them with the extensive available literature.

The English grammar, style and syntax used in the manuscript were poor and need to be completely revised.

Comments on the Quality of English Language

The English grammar, style and syntax used in the manuscript were poor and need to be completely revised.

Author Response

Dear reviewer, thank you for reading our article deeply and carefully.

Below you can find our answers:

Comments and Suggestions for Authors

The manuscript presented by Shvachko et al., aimed to identify genomic regions associated with traits of agronomic interest in a panel of bread wheat genotypes. Therefore, the subject of the paper is relevant, it falls in the general scope of the journal and could be of interest to a wide range of researchers engaged in wheat genetic studies and/or varietal development. However, in this version, the study cannot be published in a high-impact scientific journal.

The title of the manuscript suggested an analysis of the effect of different breeding periods and programs (Russia and Germany) which were not discussed.

The main idea to use cultivars released in different periods (namely that released before and after the “green revolution”) was rather to compose a GWAS panel as diverse as possible, than to focus on comparison of old and modern varieties. As for different breeding programs we have chosen Russian and German varieties based on previous data on divergence between of old and modern spring wheat varieties (namely, that related with spike productivity and grain quality) released in these countries (Hagel et al., 2005; Morozova et al., 2016, etc). In the revised MS we do not focus anymore in the article title on different breeding programs, however, we explain our considerations when compiling the GWAS panel. In electronic supplementary materials for the revised MS we included lists of varieties with low and high values for different spike productivity and grain quality traits, from which one can see that for some traits, varieties released in different periods contribute to subgroups with either highest or lowest values. We think, such way to choose contrast genotypes may be one of the reason we were able to find some new genomic loci significantly associated with evaluated traits. We discuss this issue in the revised MS.

The Abstract in this version did not adequately summarize the manuscript; too generic, and without any result.

- We corrected the abstract

Abstract: Genome-wide association studies (GWAS) are among the genetic tools for the mining of genomic loci associated with useful agronomic traits. The study enabled us to find new genetic markers associated with grain yield as well as quality. The sample under study consisted of spring wheat cultivars developed in different decades of the last century. A panel of 186 accessions was evaluated at VIR’s experiment station in Pushkin across a 3-year period of field trials. In total, 24 SNPs associated with six productivity characters were revealed. Along with detecting significant markers for each year of the field study, meta-analyses were conducted. Loci associated with useful yield-related agronomic characters were detected on chromosomes 4А, 5А, 6А, 6В, and 7В. In addition to previously described regions, novel loci associated with grain yield and quality were identified during the study. We presume that utilization of contrast cultivars originated in different breeding periods allowed us to identify new markers associated with useful agronomic characters.

In the introduction section, the issue addressed by the study was insufficiently exposed. It is important to clearly define and appropriately frame the study question.

- We have revised Introduction according this comment (please, see the revised MS)

The experimental methodology used was not appropriate to support the hypothesis proposed by the authors and, the methods described are insufficient to repeat the results. GWAS analysis should be conducted with BLUP data, to account for genotype-by-year interaction.

- Method BLUP was originally developed in animal breeding for estimation of breeding value. However, it does not have gained the same popularity in plant breeding and variety testing as it has in animal breeding. BLUP is the most widely used for genomic selection. That is, in genomic selection, to predict the phenotype from the available genotype data, a training population is first used, for which the data of both the trait under study and the full-genome genotype of individuals are known.  In this study, we searched for associations between phenotype and genotype (marker and trait). We used the Mixed liner model, which allows us to take into account the population structure of the sample by pairwise assessment of genetic similarity between the studied genotypes. Linear mixed models avoid type I errors in the case of related structure in the sample and minimize the number of false positive results [Amos C.I., 1989. Amos C.I., Elston R.C., Wilson A.F., Bailey‐Wilson J.E. A more powerful robust sib-pair test of linkage for quantitative traits. Genetic epidemiology, 1989, 6, 435–449].

The authors implied that they had conducted a meta-QTL analysis, however it is not well understood how it was conducted. No mention was made in the materials and methods section.

- We have added meta-analysis in the “Materials and methods” section:

4.8. Meta-analysis

A meta-analysis was used to enhance the statistical power of the GWAS results. [68]. Therefore, we also combined the p-values of three years for each studied character. We used modified Fisher’s method accounting for the non-independence implemented in the R package “poolr”, because the results were not independent (each year we took the same genotypes) [69]. It exploits the effective rather than nominal number of tests, thus avoiding overestimation of statistical significance. The effective number of tests was calculated using the Nyholt method [70].

The methodology and protocols for determining phenotypic traits were not detailed.

- We have added reference to detailed methodology used (published earlier) in the revised MS: Phenotyping was carried out according to the standard VIR method. (Merezhko, A.F.; Udachin, R.A.; Zuev, E.V.; Filatenko, A.A. Serbin, A.A.; Lyapunova, O.A.; Kosov, V.Yu.; Kurkiev, U.K.; Okhotnikova, T.V.; Navruzbekov, N.A.; Boguslavsky, R.L.; Abdulaeva, A.K.; Chikida, N.N.; Mitrofanova, O.P.; Potokina, S.A. Replenishment, preservation and study of the world's collection of wheat, aegilops and triticale. (Methodological instructions). St. Petersburg, VIR. 1999, 81) Usually standard techniques are not described in detail

The phenotypic traits considered in the study were not uniquely identified. In the text "Number of spikelets in a spike, SN" is confusing with "grain number in a spike, SN"; or thousand grain weight” with "grain weight per spike".

- 5. Conclusion (corrected)

The results of the field study (phenotyping) of 186 spring wheat accessions, conducted in 2021-2023 in two-fold repetition at the research and production facility of VIR, were summarized. The accessions were evaluated according to yield-related characters, such as plant height (PL), resistance to lodging (RL), spike length (SL) and 1000 grain weight (TGW). Loci associated with useful agronomic characters were detected through GWAS. The results of the study highlight the importance of conserving the gene pool of landraces as a source of valuable genetic diversity for breeding and genetic research.

number of spikelets in a spike – corrected - number of spikelets per spike

The results were not clearly explained, and in some cases, they were not preceded by an appropriate discussion in the methods section (i.e. meta-analysis).

- We have added a meta-analysis method description in the “Materials and methods” section and revised text according this comment.

In the discussion section, the authors should discuss in detail the original results (if any) obtained from the GWAS, comparing them with the extensive available literature.

- We improved the "Discussion" section according this comment.

Comments on the Quality of English Language

The English grammar, style and syntax used in the manuscript were poor and need to be completely revised.

-We improved English grammar.

Two edits should be made to the manuscript:

  • more detailed part of the results 2.2. Analysis of genotypic data (not scale balanced compared to other parts);

- We have added more detailed Results in electronic supplementary materials and also moved some figures from the main text to supplementary materials in order to make the “Results” section in the main text more balanced.

2) edit the Reference section, i.e. unify according to journal requirements.

- We have corrected this in the revised MS

Reviewer 3 Report

Comments and Suggestions for Authors

Dear authors

Happy day

The paper is informative, well designed, good selection for the tools, protocols and the statistical analysis.

Meanwhile, the paper is fine but need some improvement

1-      Kindly, use a shorter title.

2-      Kindly, avoid use over-estimated words like highly, high, many etc.

3-      Kindly, add more details in the header of the Tables.

4-      Kindly, made some of your long sentences shorter.

5-      Kindly, improve the image number 2, it might be suggested to reduce their numbers and move the rest to the supplementary data.

6-      Kindly, add references, particularly when you use kit(s) and did not describe the full protocol.

7-      Kindly, use new references (2023, 2024).

Author Response

Dear authors

Happy day

The paper is informative, well designed, good selection for the tools, protocols and the statistical analysis.

Thank you for a carefully read our manuscript. We will be happy to respond to your comments!

Meanwhile, the paper is fine but need some improvement

1-      Kindly, use a shorter title.

- The title in the revised MS is “Mining of QTLs for spring bread wheat spike productivity by comparing spring wheat cultivars released in different decades of the last century”.

2-      Kindly, avoid use over-estimated words like highly, high, many etc.

- We tried to avoid this in the revised MS

3-      Kindly, add more details in the header of the Tables.

- We have added more details in the revised MS

4-      Kindly, made some of your long sentences shorter.

- The MS text was revised to make sentences shorter

5-      Kindly, improve the image number 2, it might be suggested to reduce their numbers and move the rest to the supplementary data.

- We moved the Figure to supplementary data

6-      Kindly, add references, particularly when you use kit(s) and did not describe the full protocol.

- We have added references:

- Qiagen. BDP Handbook, 2005. Accepted manuscript online: apps.thermoscientific.com

- Merezhko, A.F.; Udachin, R.A.; Zuev, E.V.; Filatenko, A.A. Serbin, A.A.; Lyapun-ova, O.A.; Kosov, V.Yu.; Kurkiev, U.K.; Okhotnikova, T.V.; Navruzbekov, N.A.; Boguslavsky, R.L.; Abdulaeva, A.K.; Chikida, N.N.; Mitrofanova, O.P.; Potokina, S.A. Replenishment, preservation and study of the world's collection of wheat, aegilops and triticale. (Methodological instructions). St. Petersburg, VIR. 1999, 81.

7-      Kindly, use new references (2023, 2024).

- We have added references:

Li, L.; Xu, D.; Bian, Y.; Liu, B.; Zeng, J.; Xie, L.; Liu, S.; Tian, X.; Liu, J.; Xia, X.; He, Z.; Zhang, Y.; Cao, S. Fine mapping and characterization of a major QTL for plant height on chromosome 5A in wheat. Theor Appl Genet, 2023, 136. Doi: 10.1007/s00122-023-04416-9.

Leonova, I.; Kiseleva, A.; Berezhnaya, A.; Orlovskaya, O.; Salina, E. Novel Genetic Loci from Triticum timopheevii Associated with Gluten Content Revealed by GWAS in Wheat Breeding Lines. International Journal of Molecular Sciences, 2023, 24(17), 13304.

Lasky, J.; Josephs, E.; Morris, G. Genotype–environment associations to reveal the molecular basis of environmental adaptation. The Plant Cell, 2023, 35(1), 125-138.

Round 2

Reviewer 2 Report

Comments and Suggestions for Authors

The authors have satisfactorily revised the manuscript.